# Metabolic Adaptation to Sulfur of Hyperthermophilic *Palaeococcus pacificus* DY20341^T^ from Deep-Sea Hydrothermal Sediments

**DOI:** 10.3390/ijms21010368

**Published:** 2020-01-06

**Authors:** Xiang Zeng, Xiaobo Zhang, Zongze Shao

**Affiliations:** 1Key Laboratory of Marine Biogenetic Resources, Third Institute of Oceanography, Ministry of Natural Resources, No.178 Daxue Road, Xiamen 361005, China; zengxiang@tio.org.cn (X.Z.); pangdizhu342228@sina.com (X.Z.); 2School of public health, Xinjiang Medical University, No.393 Xinyi Road, Urumchi 830011, China

**Keywords:** *Palaeococcus pacificus* DY20341^T^, elemental sulfur, iron–sulfur cluster, hydrogenase, sulfur metabolism

## Abstract

The hyperthermo-piezophilic archaeon *Palaeococcus pacificus* DY20341^T^, isolated from East Pacific hydrothermal sediments, can utilize elemental sulfur as a terminal acceptor to simulate growth. To gain insight into sulfur metabolism, we performed a genomic and transcriptional analysis of *Pa. pacificus* DY20341^T^ with/without elemental sulfur as an electron acceptor. In the 2001 protein-coding sequences of the genome, transcriptomic analysis showed that 108 genes increased (by up to 75.1 fold) and 336 genes decreased (by up to 13.9 fold) in the presence of elemental sulfur. *Palaeococcus pacificus* cultured with elemental sulfur promoted the following: the induction of membrane-bound hydrogenase (MBX), NADH:polysulfide oxidoreductase (NPSOR), NAD(P)H sulfur oxidoreductase (Nsr), sulfide dehydrogenase (SuDH), connected to the sulfur-reducing process, the upregulation of iron and nickel/cobalt transfer, iron–sulfur cluster-carrying proteins (NBP35), and some iron–sulfur cluster-containing proteins (SipA, SAM, CobQ, etc.). The accumulation of metal ions might further impact on regulators, e.g., SurR and TrmB. For growth in proteinous media without elemental sulfur, cells promoted flagelin, peptide/amino acids transporters, and maltose/sugar transporters to upregulate protein and starch/sugar utilization processes and riboflavin and thiamin biosynthesis. This indicates how strain DY20341^T^ can adapt to different living conditions with/without elemental sulfur in the hydrothermal fields.

## 1. Introduction

Deep-sea hydrothermal fields provide unique and diverse habitats for various microbes. However, in such a harsh extreme environment, sharp physical and chemical gradients arise as great challenges to their survival. In addition, temporal changes with vent activity variation also pose challenges. Vent microorganisms are supposed to adapt themselves to these environmental spatial and temporal changes, helping them to survive as the chimney begins to die out or spread to vent surroundings, thereby decreasing the temperature range and necessary chemical supplies. The elemental sulfur is abundant in the mixing zones and located far from the black, high-temperature vent chimneys [1] which can be used as electron sinks in fermentation processes carried out by many Thermococcales and some representatives of Desulfurococcales [2,3]. Archaea of Thermococcales, composed of three genera (i.e., *Thermococcus*, *Pyrococcus*, and *Palaeococcus*), possess versatile metabolisms, facilitating them to grow in both hydrothermal chimnies and sediments and even the subsurface underneath hydrothermal vents [4,5,6]. All species of Thermococcales can ferment various organic compounds with S^0^ as the electron accepto, such as diverse sugars, peptides, amino acids, and organic acids. The addition of elemental sulfur enhances growth in *Thermococcus* and is essential to some strains [6,7].

Microbial versatility for energy generation may provide an ecological advantage for their survival in hydrothermal fields. The core set of genes in sulfur metabolism has been studied with *Pyrococcus furiosus* and *Thermococcus onnurineus* NA1 [8,9,10]. The reduction of elemental sulfur was found to be related to the NAD(P)H sulfur oxidoreductase–membrane hydrogenase (Nsr-MBX) S^0^-reduction system to transfer the electron flow to S^0^ and generate NADPH for Nsr [8]. Other multifunctional enzymes, including sulfide dehydrogenases (SuDHs), cytoplasmic hydrogenases (SH), pyruvate oxidoreductase, disulfide oxidoreductase (Pdo), and sulfur related transcriptional regulator (SurR) [9,10,11], were also found to be upregulated. In other words, the assimilation of sulfur also induced iron transport and iron–sulfur cluster biosynthesis. In *Pyrococcus furiosus* and *Thermococcus onnurineus* NA1, the secondary response to S^0^ includes the upregulation of genes involved in iron transport (feoB) and iron–sulfur cluster biosynthesis (sufBD) and of iron–sulfur-cluster-containing enzymes (glutamate synthase and 3-isopropylmalate dehydratase), indicating an increased need for iron–sulfur metabolism under S^0^-reducing conditions. Sulfur-responsive proteins (SipAB) and Fe–S cluster-carrier proteins, e.g., Mrp,Nbp35, may be involved in iron–sulfur metabolism [9,10]. The *Palaeococcus* genus, a deep-branching lineage of the Thermococcales, comprises only three species to date, *Palaeococcus ferrophilus* [12], *Palaeococcus helgesonii* [13], and *Palaeococcus pacificus* [14]. *Palaeococcus pacificus* DY20341^T^ (=JCM 17873^T^; =DSM 24777^T^) is a hyperthermophilic piezophilic anaerobic archaeon that was isolated from East Pacific Ocean hydrothermal sediments (S 1.37° W 102.45°) at a depth of 2737 m [14]. The genome of *Palaeococcus pacificus* was sequenced and was a single circular 1,859,370 bp chromosome without an extrachromosomal element [15]. Given the close phylogenetic relationship between the genera *Pyrococcus*, *Thermococcus*, and *Palaeococcus*, we deemed it interesting to compare the strategy of sulfur reduction in the Thermococcales. To date, the global view of the metabolic characteristics of S^0^ reduction growth in *Palaeococcus* species is still limited. Here, we analyzed the metabolisms of *Palaeococcus pacificus* based on further analysis of the genome and its transcriptome in response to sulfur with the aim of understanding its adaptation to a hydrothermal vent environment (chimney or sediments). Comparisons of the sulfur mechanisms between *Thermococcus* and *Pyrococcus* species were also discussed.

## 2. Results and Discussions

### 2.1. The Growth Curve of Pa. pacificus with or without Elemental Sulfur

In Thermococcales rich medium (TRM) without S^0^ (details in Methods), the DY20341 cell density reached 7.15 × 10^7^ cells/mL after 30 h, and the doubling time was about 200 min with hydrogen production after 5 h cultivation under 80 °C. The presence of elemental sulfur evidently enhanced growth. In TRMS medium, DY20341 exhibited better growth—up to 9.7 × 10^8^ cells/mL in 20 h with a shorter doubling time of 66 min under 80 °C. The elemental sulfur can be reduced to hydrogen sulfide at a final concentration of 1.20 mM/L after 60 hours’ growth, whereas H_2_ production was not detected (Figure 1).

### 2.2. Genomic and Transcriptomic Features of Pa. pacificus DY20341^T^

Based on the reported genome [15], further comparison of the general features of *Pa. pacificus* DY20341^T^ with other Thermococcales strains are listed in Appendix A. Their genome sizes were similar between 1.84 and 1.91 Mb, while *Pa. pacificus* had a lower proportion of protein-coding regions (78.11%, 1563/2001). Forty-eight percent (a total of 971) open reading frames (ORFs) were assigned as hypothetical proteins. This implied that *Palaeococcus* species have more unknown sequences than other Thermococcales strains. Besides, *T. onnurineus* had a higher G+C content than *Pa. pacificus* and *P. furious* (Appendix A).

To analyze the transcriptional changes specific to the sulfur response, the RNAs were collected in triplicate under two different substrate culture conditions (TRM medium and TRMS medium with sulfur). Using the statistical criteria described previously, a 2.0 log2 (ratioRPKM, Reads Per Kilobase per Million mapped reads) of median cutoff was considered as differential gene transcription under the two growth conditions [16]. A total of 454 genes showed a differential transcription profile, of which 108 increased by up to 75.1 fold and 336 decreased by up to 13.9 fold in the presence of S^0^ in TRMS medium compared to S^0^ absence control in TRM medium. 

### 2.3. Metabolic Response to Growth on Sulfur 

#### 2.3.1. Central Metabolism

The main central metabolism in *Pa. pacificus* includes the modified Embden–Meyerhof glycolytic pathway, pentose phosphate synthesis, and uncomplete tricarboxylic acid cycle (Figure 2) which were analyzed as follows. 

Glycolysis in *Pa. pacificus* is via the Embden–Meyerhof pathway: ADP-dependent glucokinase (GLK, PAP_01430), phosphoglucose isomerase (PGI, PAP_01425), ADP-dependent phosphofructokinase (PFK, PAP_04875), triosephosphate isomerase (TIM, PAP_09970), fructose-1,6-bisphosphate aldolase (FBA, PAP_03775), phosphoglycerate mutase (PGM, PAP_01310, PAP_04825), enolase (ENO, PAP_04410), pyruvate kinase (PYK, PAP_08515), and phosphoenolpyruvate synthase (PPS, PAP_01340) (Figure 2). *Palaeococcus pacificus* chooses two pathways of catalyzing glyceraldehyde-3-phosphate (G3P) metabolism: (1) the bacterial/eukaryotic classic Embden–Meyerhof (EM) glycolytic pathway including glyceraldehyde-3-phosphate dehydrogenase (GAPDH, PAP_02855), phosphoglycerate kinase (PGK, PAP_01540), and the modified Embden–Meyerhof pathway of Thermococcales archaea including glyceraldehyde-3-phosphate; (2) ferredoxin oxidoreductase (GAPOR, PAP_09040), and non-phosphorylating glyceraldehyde-3-phosphate dehydrogenase (GAPN, PAP_03390). The expression of GAPOR increased under conditions of growth in S^0^ (2.11 fold compared to without S^0^). However, GAPN was evidently downregulated (log_2_ (TRMS/TRM) = −2.41). This indicated that GAPOR (PAP_09040) was induced by S^0^ to generate ferredoxins which may be used by MBX on the membrane. Subsequently, pyruvate was oxidized to acetyl-CoA by pyruvate ferredoxin oxidoreductase (POR, PAP_04320-04330) and, finally, to acetate by acetyl-CoA synthetase (ACS, PAP_03610, PAP_05270, PAP_05870, PAP_06970). Pyruvate was also degraded to formate by pyruvate formate lyase (PAP_06440, PAP_06540). Pyruvate can also form alanine by alanine aminotransferase (Ala AT, PAP_08180). Only ACS I(PAP_03610) was upregulated, triggered by sulfur (Appendix A).

The genomic properties for pentose–phosphate metabolism and adenosine 5’-monophosphate (AMP) metabolism in *Pa. pacificus* were found (Figure 2). They included a unique type III ribulose bisophosphate carboxylase (Rubisco, PAP_06885) [17]. *Palaeococcus pacificus* harbors both homologs for AMP phosphorylase (DeoA, PAP_04670) and ribose-1, 5-bisphosphate isomerase (RBPI, PAP_04815), enzymes required to supply the RuBisCO substrate, ribulose-1,5-bisphosphate, from AMP and phosphate. In this pathway, adenine is released from AMP and the phosphoribose moiety enters a central-carbon metabolism [17]. No phosphoribulokinase and acetyl coenzyme A synthase—key enzymes in the Calvin cycle and Wood–Ljungdahl pathway—were detected, as in any of the Thermococcales genomes. The pentose–phosphate metabolism and AMP metabolism had no differential expression in the presence of S^0^. No CO dehydrogenase, a central enzyme in microbial carbon monooxide (CO) metabolism, was found in the genome of *Pa. pacificus*. This was different than *T. onnurineus* NA1 [10]. 

The TCA cycle not only produces NADH for ATP synthesis via the electron transport chain, but also plays a key role in the synthesis of intermediates for anabolic pathways; specifically, 2-ketoglutarate, oxaloacetate, and succinyl-CoA are starting points for the synthesis of glutamate, aspartate, and porphyrin, respectively [18]. *Palaeococcus pacificus* possesses an incomplete tricarboxylic acid (TCA) cycle, similar as most Thermococcales. *Palaeococcus pacificus* only contains succinyl-CoA synthase (SCS, PAP_00035, PAP_05875), alpha and beta subunits of fumarate hydratase (PAP_05860 and PAP_05865) and α-Ketoglutarate ferredoxin (Fd) oxidoreductases (KGOR, PAP_05645-05675) (Figure 2). Otherwise, phosphoenolpyruvate (PEP) carboxylase (PC, PAP_00835), PEP carboxykinase (PCK, PAP_02550), and pyruvate carboxylase (PVC, PAP_03770) carry out aplerotic reactions that produce oxaloacetate as a TCA cycle intermediate. 

These results indicate that there was no obvious differential expression in the central pathway in the two different cultures (i.e., TRMS medium and TRM medium). Notably, in the second glyceraldehyde-3-phosphate (G3P) metabolism pathway, GAPOR (PAP_09040) was induced by S^0^ to generate ferredoxins for membrane hydorgense MBX, which helps in ATP generation.

#### 2.3.2. Utilization of Proteins

Thermococcales species can use peptides as their sole carbon and energy sources with proteolytic activities. Most of them have diverse and abundant protease and peptidase, for which the optimal temperature range is 65–115 °C [19]. In order to assimilate the proteinous substrates, the *Pa. pacificus* genome encodes diverse proteases, a peptide transport system, and an amino acid degradation pathway (Figure 2).

The genome of *Pa. pacificus* includes an extracellular subtilisin-like protease (PAP_01675), membrane-bound protease (PAP_01420, PAP_02280), intracellular protease (PAP_03090, PAP_05230, PAP_05570, PAP_09635), and a membrane-bound aminopeptidase (PAP_00355). Half of the encoded proteases (6/11) in *Pa. pacificus* were downregulated, while one extracellular protease was upregulated upon growth with elemental sulfur (Appendix A). Three ORFs (PAP_02280, PAP_04645-04650) for post-translational modification were found to be downregulated in the TRMS culture. The crude extracts from culture supernatant or cell sonicate were assayed for proteolytic activity using casein as a substrate at 60 °C, 80 °C, and 100 °C. The result showed that *Pa. pacificus* can produce thermophilic protease—the supernatants which have higher activities (15.1–22.0 U/mL) than sonicates (0.1–4.5 U/mL) (Appendix A). Compared with the two culture conditions (i.e., TRM and TRMS), S^0^ may trigger some extracelluar proteases which have a lower optimal temperature (60 °C). In accordance with the transcriptome result, it may be an extracellular protease (PAP_01675). Cell sonicate of TRMS media had a slightly lower activity than TRM, while both the intracellular activities were low. Most downregulated intracellular proteases have low RPKM values and play a role in post-translational modification. The protease activity assay with casein as a substrate may limit their analysis. Therefore, further quantitative analysis is necessary.

The cell can import peptides by an ABC-type dipeptide/oligopeptide transport system (PAP_00150-00170; PAP_06335-06355) and oligopeptide transporter (PAP_10185-10210). The imported peptides can be cleaved by 15 encoded peptidases. The amino acids can then be deaminated by a number of aminotransferases (Appendix A) in a glutamate dehydrogenase (PAP_09695)-coupled manner, followed by oxidation to generate the corresponding coenzyme A (CoA) derivatives and aldehydes as proposed for *P. furiosus* and *T. sibiricus* [20]. Notably, the ABC-type dipeptide/oligopeptide transport system II (PAP_06335-06355) is closest with the bacterial strain *Aciduliprofundum boonei* T469. We found this cluster only exists in the genomes of certain Thermococcales strains (18/40).

Interestingly, with sulfur in the medium, dipeptide transport system (PAP_00150-00170; PAP_06335-06355), branched-chain amino acid ABC transporter (LivKGFHM, PAP_03900-03925), and some amino acid permease (PAP_03785; PAP_05545; PAP_06890; PAP_06940; PAP_09570) were downregulated (Appendix A). The results showed that cells decreased the active transport of peptides and amino acids.

The oxidation of amino acids involves at least four types of ferredoxin-dependent oxidoreductases with distinct substrate specificities: pyruvate:ferredoxin oxidoreductase (POR, PAP_04320-04330); 2-ketoisovalerate:ferredoxin oxidoreductase (VOR, PAP_04305-04315); indolepyruvate:ferredoxin oxidoreductase (IOR, PAP_05275-05280); 2-oxoglutarate oxidoreductase (OGOR, PAP_05645-05675). The acyl-CoA derivatives in *Pa. pacificus* are converted to the corresponding acids and succinate by the acetyl-CoA synthetases (ACS, PAP_03610, PAP_05270, PAP_05870, PAP_06970) and a succinyl-CoA synthetase (SCS, PAP_00035, PAP_05875) with concomitant substrate-level phosphorylation to generate ATP, the same as shown in *P. furiosus*, *T. kodakaraensis*, and *T. sibiricus* [20,21].

As an alternative assimilation pathway for amino acids, depending on the redox balance of the cell, 2-oxo acids derived from amino acids can be decarboxylated to corresponding aldehydes by ferredoxin-independent reactions of the ferredoxin-dependent oxidoreductases (OFOR, PAP_03305, PAP_03310) and then oxidized to carboxylic acids by the function of aldehyde:ferredoxin oxidoreductases (AOR, PAP_09030) and formaldehyde:ferredoxin oxidoreductase (FOR, PAP_08640). Alcohol dehydrogenases (ADH, PAP_03980) might be responsible for the reduction of aldehydes to alcohols in the absence of sufficient amounts of the terminal electron acceptor. S^0^, in the reaction coupled to the oxidation of NADPH to NADP, would dispose of excess reductant [20]. In this condition, ferredoxin-dependent oxidoreductases (OFOR, PAP_03305, PAP_03310) were activated. (Appendix A). 

According to the genomic and experimental analysis, *Pa. pacificus* can produce thermophilic extracellular and intracellular proteases. Some peptide and amino acid transports were found decreased. Depending on the excess reductant by sulfur reducing, it may choose the alternative assimilation pathway, for example, ferredoxin-dependent oxidoreductases were activated, and some alternative oxireductase (FOR, PAP_08640; IOR, PAP_05280) were downregulated.

#### 2.3.3. Utilization of Carbohydrates

*Palaeococcus pacificus* was found to be able to grow on starch. Extracellular α-amylase (PAP_00275) and intracellular α-amylase (PAP_09095) belong to the family GH13 which can hydrolyse alpha bonds of large, alpha-linked polysaccharides such as starch and glycogen, yielding glucose and maltose. Extracellular cyclodextrin glucosyltransferase (PAP_01075), belonging to the GH13 family, could synthesize non-reducing cyclic dextrins, known as cyclodextrins, starting from starch, amylose, and other polysaccharides. 4-α-Glucanotransferase (PAP_09225)—affiliated to the glycoside hydrolase family GH-57—transfers a segment of a 1,4-alpha-D-glucan to a new position in an acceptor carbohydrate which may be glucose or a 1,4-alpha-D-glucan. Extracelluar cyclodextrin glucanotransferase (PAP_01075) of *Pa. pacificus* catalyzes the production of, predominantly, alpha-cyclodextrin (CD) from starch (Figure 2, Appendix A). The crude amylase of *Pa. pacificus* showed good stability at the temperature range from 60–100 °C.

The “maltose and trehalose degradation gene” island is absent in *Pa. pacificus* but present in *T. sibiricus* and *P. furiosus* [20]. However, α-galactosidase (PAP_04140), β-galactosidase (PAP_04145), trehalose synthase (PAP_05120), glycosidase (PAP_08185), glycogen-debranching enzyme (PAP_08190), and amylopullulanase (PAP_05485) were found in the genome of *Pa. pacificus*. These genes encode enzymes potentially responsible for maltose and trehalose degradation.

Furthermore, five carbohydrate ABC-type transport systems were identified in the genome of *Pa. pacificus* to transport short and longer oligomers. The maltotriose and longer oligomers, including starch, can be transported by Mal-I (PAP_04995-05020), Mal-IV (PAP_05160-05175), and Mal-EFGK; Mal-II (PAP_05100-05110), closest to the Mal-II transporter of *T. sibiricus* [20], is specific for maltooligosaccharides; Mal-III (PAP_05125-05135) recognizes and transports maltose and trehalose; the sugar ABC-type transport system (PAP_05160-05175) can transport sugar, e.g., N-acetyl-D-glucosamine and maltose (Figure 2, Appendix A).

When elemental sulfur was used as an electron acceptor, galactosidase, glycosidase, Mal-II, Mal-IV, and sugar transporters were downregulated. As a result, sulfur reduction may generate energy to compensate for the energy utilization of carbohydrates.

In addition, glycerol can be transported into the cell by glycerol transport protein (PAP_03945), phosphorylated by glycerol kinase (PAP_02595-02605; PAP_08525, PAP_08540) and, then, enter the modified EM pathway. In the presence of S^0^, the pathway of glycerol for glycogenesis was evidently downregulated (Appendix A). 

Different to *T. sibiricus* and *Pyrococcus* species [20], *Pa. pacificus* does not contain a “saccharolytic gene island” which carries a set of genes responsible for the utilization of cellulose, laminarin, agar, and other β-linked polysaccharides. These genes are also absent in other *Thermococcus* species. This agrees with the inability of this archaeon to grow on cellobiose or longer β-linked glucans.

All gene clusters related to chitin utilization are absent in *Pa. pacificus*, different to *Pyrococcus* and *Thermococcus*, such as *P. furious* [22], *P. chitonophagus* [23] and *T. kodakaraensis* KOD [24]. The lipolytic growth of *T. sibiricus* appears to be unique in the order Thermococcales [20].

*Palaeococcus pacificus* prefers starch other than cellulose, chitin, and lipids, which is different to *Pyrococcus* and *Thermococcus*. Some starch utilizing pathways were downregulated in the sulfur culture which may be caused by the inactivation of the regulator TrmB (PAP_08070) (See Section 2.3.6.)

#### 2.3.4. Energy Metabolism

The conserved respiration system is represented by membrane-bound hydrogenase and A_0_A_1_-type ATP synthase, which are highly similar to the gene clusters in available Thermococcales genomes. This suggests the ancestral state of the gene clusters in the Thermococcales and the central role they play in these organisms [25,26]. Two adjacent copies of the MBH cluster (MBH1, PAP_01095- 01160; MBH2, PAP_01165-01230) were identified in *Pa. pacificus (*Figure 2 and Figure 3) as well as in *Thermococcus* sp. 4557, *T. sibiricus*, and *T. barophilus* [20,27,28]. One membrane-bound NADP-reducing hydrogenase complex (MBX, PAP_02355-PAP_02415), eventually linked to the reduction of S^0^, was also identified in the genome of *Pa. pacificus* (Figure 3). The MBX complex has Fe–S binding motifs similar to those of MBH, indicating that it can act as an electron acceptor from ferredoxin and reduces NADP to NADPH, allowing Nsr (PF1186) to generate H_2_S [8]. The genes encoding the components of the MBX complex increased 2.29–8.76 folds in the presence of S^0^ (Appendix A). Hydrogenase maturation proteins were obviously upregulated, especially metal ion binding protein HypF (PAP_01595) which was upregulated by 10.80 fold compared to those without sulfur (Table 1). It may contribute to producing MBX hydrogenase. NAD(P)H sulfur oxidoreductase (Nsr, PAP_01275) also increased 2.4 fold to generate H_2_S. It was found that sulfide dehydrogenases (SuDHI:PF1327-1328, TON_0057; SuDHII:PF1910-1911 TON_1336-1337) were strongly upregulated during growth on sulfur [10,29]. As in *P. furious* [8], *Pa. pacificus* DY20341 contains only one sulfide dehydrogenase (SuDH, PAP_00200-00205) with strong upregulation (log2(TRMS/TRM) = 1.73; 2.19). In *T. kodakarensis* KOD1 and *T. onnurineus* NA1, NADH:polysulfide oxidoreductase (NPSOR) is capable of generating H_2_S to Nsr [10,26]. The homolog of NADH: polysulfide oxidoreductase (NPSOR) was found in *Pa. pacificus* DY20341(PAP_03325), *Pa. ferrophilus* (PFER_RS00565) and *P. furious* (PF1197) but not in other *Pyrococcus* species. A previous study found that deletion of Nsr did not show obvious growth defects on S^0^ and only deletion of MBXL (the catalytic subunit of the MBX complex) significantly reduced the growth of the archaeon [9]. It underlined that there may exist another oxidoreductase to substitute Nsr in *P. furious* and *Pa. pacificus*. When in the presence of S^0^, the MBH1 complex was slightly upregulated, whereas MBH2 was slightly downregulated. Furthermore, two cytoplasmic (Ni–Fe) hydrogenases (encoded by PAP_01490-PAP_01505, PAP_03240-PAP_03255; Figure 1 and Figure 3) were homologous to the SHI and SHII of *P. furiosus* DSM3638, Hyh-I and Hyh-II of *T. kodakaraensis* KOD1, and the Sulf-I and Sulf-II cluster of *T. onnurineus* NA1. These enzymes may function as an H_2_ uptake hydrogenase during H_2_ production [30,31]. In DY20341, only SHI (PAP_01490-PAP_01505) was obviously downregulated by 2.66–3.20 fold with sulfur as an electron acceptor (Appendix A). Whereas in the *P. furious* DSM3638, the addition of S^0^ induced the downregulation of all hydrogenase-related genes, except for MBX, encoding MBH and two cytosolic hydrogenases SHI and SHII (Table 2).

The exportation of Na^+^/H^+^ by MBH1, MBH2, and MBX is required to convert the proton gradient into a Na^+^ ion gradient used for ATP synthesis which is conducted by the V-type ATP synthase (PAP_09395-PAP_09435) with no obviously different expression with S^0^, different to what was observed with *P. furious* and *T. onnurineus* NA1 (Table 2 and Appendix A). 

Formate dehydrogenases subunit alpha (FdhA, PAP_03225) and formate transporter (PAP_03230) were adjacent to (Ni–Fe) hydrogenases which may be attributable to re-oxidation of formate rather than to formate production as described in *Archaeoglobus fulgidus* and *T. onnurineus* NA1 [31,32]. Different from in *T. onnurineus* [10], this process was down–upregulated in *Pa. pacificus.*

The NADH:ubiquinone oxidoreductase (complex I, PAP_01510-PAP_01525) (EC:1.6.5.3)—a respiratory-chain enzyme that catalyses the transfer of two electrons from NADH to ubiquinone in a reaction that is associated with proton translocation across the membrane—was slightly downregulated. Only ubiquinone/menaquinone biosynthesis methyltransferase (PAP_08725) was found.

#### 2.3.5. Sulfur Assimilation

Except as an energy supply, the main functions of sulfur in the cell include synthesis of cofactors (molybdenum cofactor, iron–sulfur clusters), sulfuration of tRNA, modulation of enzyme activities, and regulating the redox environment [33]. 

In the three key enzymatic steps of the sulfate reduction pathway, only sulfate adenylyltransferase (sat, PAP_09885), adenylylsulfate kinase (cysC, PAP_09900), and PAPS reductase (cysH, PAP_04545) were found which reduce the activated sulphate to adenylsulfate (APS) and 3’-phosphoadenylyl sufate (PAPS) to sulfite (Figure 2, Appendix A). No adenosine-5’-phosphosulfate reductase (apr) or dissimilatory (bi)sulfite reductase (dsr) were found. Only sulfate adenylyltransferase (sat) was obviously upregulated in TRMS medium (1.97 fold compared to TRM medium). This indicates that the amount of APS and sulfite in vivo may be accelerated.

Meanwhile, the biosynthesis of cysteine was reduced in sulfur-supplied media. Cysteine synthase (PAP_05900), cystathionine beta-synthase (PAP_08745), and cystathionine gamma-lyase (PAP_08750) were found to be obviously down regulated by 2.36–7.25 fold in the presence of sulfur. Another way to produce cysteine is primarily on tRNA-Cys via the SepRS/SepCysS pathway [34]. Cysteinyl-tRNA synthetase (PAP_06715) had no differential expression in sulfur-supplemented media. The homologs of cysteine desulfurases were lacking in DY20341 which transfers cysteine to the Fe–S cluster assembly protein. A possible reason was that exogenous sulfide instead of cysteine was the dominant sulfur source for the iron–sulfur cluster which was formerly found in the methanogenic archaeon *Methanococcus maripaludis* [35].

The protein disulfide oxidoreductase (PDO:PF0094, TON_0319), a SurR-regulated glutaredoxin-like protein involved in the maintenance of protein disulfide bonds, was upregulated during growth on sulfur in *T. onnurineus* NA1 [10]. But, the homolog (PAP_07225) of PDO in *Pa. pacificus* was found without upregulation upon the addition of sulfur.

Besides, a previous study found that sulfur assimilation metabolism was strongly associated with iron [8]. In *P. furiosus*, the secondary response to S^0^ includes the upregulation of genes involved in iron transport (feoB) and iron–sulfur cluster biosynthesis (sufBD) and of iron–sulfur-cluster-containing enzymes (glutamate synthase and 3-isopropylmalate dehydratase), indicating an increased need for iron–sulfur metabolism under S^0^-reducing conditions and that SipA (PF2025) may be involved in iron–sulfur metabolism [9]. The elemental sulfur-responsive protein (sipA) is regulated by sulfide in an iron-dependent manner and is proposed to play a role in intracellular iron sulfide detoxification [36]. In *T. onnurineus*, the encoding genes of iron–sulfur cluster biogenesis were also upregulated including SipA (TON_0919), SipB (TON_0916), SufC (TON_0530), SufBD-related proteins (TON_0531 and TON_0849-0850), and the Fe–S cluster carrier protein Mrp/Nbp35 family ATP-binding protein (TON_1843) [10]. 

In *Pa. pacificus*, ABC-type iron (III)-siderophore transporter (PAP_06035-06040) and ferrous iron transporter (feoAB, PAP_06060-06065) were highly upregulated under sulfur culture conditions (Figure 2, Appendix A). Whereas, ABC-type iron (III) transporter (PAP_03965-PAP_03970) was downregulated. The homolog of SipA (PAP_06520, Table 1), which is an iron–sulfur cluster binding protein was obviously upregulated with log2 (TRMS/TRM) = 6.23. No homolog of SipB (PF2026) was found in *Pa. pacificus* (Table 2). Some iron–sulfur cluster assembly proteins (IscSU, PAP_08965, PAP_08975) and the sulfur formation system (Suf SCB, PAP_01245, PAP_02345, PAP_02350) were found in *Pa. pacificus*, whereas they were slightly downregulated (log2 (TRMS/TRM) value = −0.15 to −0.93) in sulfur-containing medium. Many other iron–sulfur cluster binding proteins were upregulated. Cobalamin biosynthesis protein CobQ and 4Fe-4S ferredoxin type protein (PAP_06425-06430, Table 1) were highly upregulated (log2 (TRMS/TRM) values were 3.74 and 3.41, respectively). Radical SAM protein (PAP_00325-00335), an Fe–S cluster-containing enzyme, was upregulated (log2 (TRMS/TRM) value of 1.61 with a high number of reads (36622/11960)). The Mrp/NBP35 ATP-binding protein (PAP_09000), which functions as an iron–sulfur cluster carrier [37], was obviously upregulated (log2 (TRMS/TRM) value was 2.18). We concluded that the accumulation of S^0^ will promote iron transfer to intracellular proteins and then further synthesize more Fe–S proteins. These Fe–S clusters are used as “molecular switches” for gene regulation at both the transcriptional and translational levels due to the fact of their sensitivity to cellular redox conditions.

Riboflavin and thiamin are the co-enzymes in numerous oxidation and reduction reactions. In the S^0^-supplied culture, the genes involved in the biosynthesis of riboflavin (PAP_00630-00600) and thiamin (PAP_03985-03995) were both obviously downregulated (Appendix A). Purine nucleotides are essential metabolites for the biosynthesis of riboflavin. With sulfur as an extra electron donor, purine biosynthesis was also obviously downregulated (PAP_00540-00605). Compared with highly upregulated cobalamin biosynthesis protein, this implies that a different cofactor was used to switch different catalytic centers under environments with/without S^0^. 

In the sulfur supplementary culture, more sulfite was produced by sulfur-reducing pathways than dissimilatory sulfate-reduction pathways, and further directly synthesized to more H_2_S and iron–sulur clusters than cystein. 

#### 2.3.6. Transcription Regulation under Sulfur as an Electron Acceptor 

In the genome of *Pa. pacificus,* a total number of 13 putative transcription regulators can be found. Of them, five were found downregulated and three were found upregulated in response ro sulfur (Appendix A). In *P. furiosus*, two transcriptional regulators (SurR, PF0095, PF2051-2052) containing an ArsR-type DNA-binding domain were found to downregulate membrane hydrogenase (MBH) [8,38]. Four ArsR-type transcriptional regulators were found in *Pa. pacificus* DY20341 (PAP_08835, PAP_06395, PAP_03045, PAP_07220, Appendix A). Three of them were obviously downregulated (log2 (TRMS/TRM) value = −2.15, −1.30, −1.25) with the addition of sulfur with only PAP_07220 not being found to be evidently downregulated. The ArsR subfamily of helix-turn-helix bacterial transcription regulatory proteins (winged helix topology) includes several proteins that appear to dissociate from DNA in the presence of metal ions [39]. In addition, transcription termination protein NusA (PAP_02730), XRE family transcriptional regulator (PAP_09615), and CopG transcriptional regulator (PAP_09880) were differentially upregulated in response to sulfur. The TrmB transcription regulator (PAP_08070) and AsnC-type transcription regulator (PAP_05565) were downregulated in contrast to their upregulation in *T. onnurineus* NA1 [10]. In *T. litoralis* and *P. furiosus*, TrmB is a dimeric protein that regulates the expression of an operon encoding trehalose/maltose transporters (TM operon) [40,41].

#### 2.3.7. Cell Mobility

*Palaeococcus pacificus* DY20341 has a complete operon flaABCDGHIJ (PAP_04440-04485). Interestingly, all the flagella operons were significantly downregulated in the presence of sulfur (Figure 2, Appendix A). Enzymes (WecB, WecC, PAP_07020-07025) that are involved in the biosynthesis of UDP-acetamido sugars (precursors for N-linked glycosylation of flagellin) [10] showed no significant differential expression. This result implies that, in the absence of sulfur, mobility is important for obtaining more proteinous and carbon substrates to swim with the flagella. 

#### 2.3.8. Mobile Elements (Transporters, Viruses, and CRISPR Elements) 

Repetitive insertion sequences (IS) elements have been proposed to be very useful to differentiate hyperthermophilic strains and to obtain additional insights into their phylogeny [42,43]. Three types of homologous *P. furiosus*-insertion sequences, IS-pfu-I, IS-pfu-II, and IS-pfu-III, were found abundantly in the complete genome sequence of *P. furiosus* DSM 3638 (GenBank accession number: AE009950) and other related *Pyrococcus* species isolated from the Mediterranean Sea [42,44]. The *T. kodakaraensis* genome contains seven genes for probable transposases and four virus-related regions. Seven copies of putative transposons assigned to three types were detected in the genome of *T. sibiricus* MM739. The *T. gammatolerans* genome has an absence of genes encoding transposase but carries two virus-related regions. Analysis of the repeated sequences and a search against the IS database revealed that a remarkable feature of the *Pa. pacificus* genome is the absence of mobile genetic elements including transposons, transposases, integrases, and virus-related regions. It indicates that the *Pa. pacificus* genome has not been subjected to frequent genomic rearrangements during its evolution. *Palaeococcus pacificus* may survive in lower temperature habitats compared to other members of Thermococcales which could be attributed to the loss of mobile elements.

The CRISPR system and restriction modification systems are types of protection against infection by phages and other sources of foreign DNA. Three CRISPRs were identified in the genome of *Pa. pacificus*. Fifty-one repeat spacer units (735977-739414 coordinates) were included in Pap-crisper-1. Twenty-six repeat spacer units (1003934-1005710) were assigned to Pap-crisper-2. The Pap-crisper-3 contains a Type I CRISPER-Cas system which includes 31 repeat spacer units (1426423-1428518) and Cas 1, Cas 2, Cas 3, Cas 4a, and Cas 6. Other representatives of the order Thermococcales generally harbor 1–10 CRISPR loci (data from CRISPERdb, http://crispr.u-psud.fr/crispr/). When these sequences are transcribed and precisely processed into small RNAs, they guide a multifunctional protein complex (Cas proteins) to recognize and cleave invading foreign genetic materials. A type I restriction–modification system was encoded in the genome of *Pa. pacificus*, including Subunit M of DNA-methyltransferase (PAP_05190), Subunit S (PAP_05210) and Subunit R of site-specific deoxyribonuclease (PAP_05225). Type II restriction endonuclease (PAP_04610) was also identified. The CRISPR-associated proteins (Cas1, Cas2, Cas4a; PAP_07940-PAP_07950) were found to be downregulated in the presence of sulfur (Appendix A). It may imply that the risk of infection is lower in sulfur enrich environments.

#### 2.3.9. Adaptation against High Temperature, High Pressure, and Oxygen 

Adaptation to high-temperature conditions is mainly correlated to the production of more stable proteins [45]. In hyperthermophilic archaea, it has been proposed that the small heat-shock protein, prefoldin, and FK506-binding protein (FKBP) function as chaperones in repressing the aggregation of unfolded proteins, followed by refolding with group II chaperonin (Hsp60 homolog) [26]. The *Pa. pacificus* genome lacks the three largest MW chaperones (Hsp 70, Hsp 90, and Hsp 100) and shares Hsp20 (small heat-shock protein, PAP_04835, PAP_07685, PAP_08375), Hsp60 chaperonine group II (thermosome, PAP_00520, PAP_02850), α- (PAP_04755) and β-subunits (PAP_02155) of prefoldin and archaeal heat-shock regulator, and ArsR family (PAP_03045, PAP_04990, PAP_05280, PAP_06790, PAP_07220, PAP_08070). Among them, four heat-shock regulators of ArsR were obviously downregulated (Appendix A) which might have overlapping functions in terms of sulfur-reducing regulatory processes as those in SurR. Thermostability can also be achieved by a reduced frequency of the thermostable amino acids such as histidine, glutamine, and threonine and an increased number of both positively charged and negatively charged residues which suggests that ionic bonds among oppositely charged residues may help to stabilize the protein structure at high temperatures [46]. In terms of the amino acid frequency of the DY20341 genome, glutamine has the second highest frequency (8.87% mol/mol), whereas histidine and threonine only constitute 1.62% mol/mol and 4.33% mol/mol.

Piezophily, in terms of the physicochemical properties of amino acids and genetic code, was also ranked by PAI (pressure asymmetry index) after comparison of the proteins from *P. furiosus* and *P. abyssi* as in the work by Massimo Di Giulio [47]. The pattern of asymmetries in the amino acid substitution process identifies the amino acids arginine, serine, glycine, valine, and aspartic acid as those having the most barophilic behavior and tyrosine and glutamine as the least barophilic. In the genome of *P. yayanosii* CH1, alanine and arginine have higher PAI values and ratio of appearances than those in other *Pyrococcus* species [48]. Hydrophobic and non-polar amino acids have a higher discriminative ability for the thermophilic-piezophilic groups [49]. Our data show that the amino acids leucine (10.43% mol/mol), glutamine (8.87% mol/mol), isoleucine (8.20% mol/mol), lysine (8.20% mol/mol) are the most prevalent amino acids in *Pa. pacificus* DY20341. This is not the case in *P. abyssi* and *P. yayanosii* CH1. This result suggests that there may be no specific piezophilic or thermophilic amino acids in some thermo-piezophiles such as *Palaeococcus*.

To cope with oxygen stress, anaerobic microorganisms can detoxify the superoxide anion by superoxide reductase (Sor) in place of superoxide dismutase in aerobic organisms [50]. The highly conserved cluster with rubrerythrin (PAP_03320; PAP_03360; PAP_03370; PAP_03375; PAP_03385; PAP_03425), rubredoxin (PAP_03330), and desulfoferrodoxin (PAP_03335) were also found in the *Pa. pacificus.* Different from *T. onnurineus* NA1, the superoxide-reducing system did not strongly downregulate proteins in *Pa. pacificus* (Appendix A). No NADH oxidases and peroxiredoxins were identified in the genome of *Pa. pacificus.* One osmoprotein cluster (PAP_05625-05640) was found in the genome of *Pa. pacificus* which may transfer from *Archaeoglobus fulgidus* DSM 4304 due to the high similarity among these two homologs. It plays a role in osmoregulation of *Pa. pacificus* during salt response.

In the presence of sulfur in the medium, both a DNA repair and recombination protein (RadB, PAP_06210) and a restriction endonuclease (PAP_06965) were downregulated (Appendix A). High expression of restriction–modification system genes can result in the death of a microbial host cell because of endonuclease cleavage of the host DNA. Their high expression may result in the low cell number when DY20341 was grown in TRM medium.

## 3. Materials and Methods 

### 3.1. Bacterial Strains and Growth Conditions

*Palaeococcus pacificus* DY20341^T^ (=JCM 17873^T^; = DSM 24777^T^) was isolated from the sediment sample collected from East Pacific Ocean hydrothermal sediments. It was grown anaerobically on a Thermococcales-rich medium (TRM) that had the following composition (per litre distilled water): 3.3 g 2,2′-piperazine-1,4-diylbisethanesulfonic acid (PIPES) disodium salt, 23 g NaCl, 5 g MgCl_2_. 6H_2_O, 0.7 g KCl, 0.5 g (NH_4_)_2_SO_4_, 1 mL KH_2_PO_4_ 5%, 1 mL K_2_HPO_4_ 5%, 1 mL CaCl_2_.2H_2_O 2%, 0.05 g NaBr, 0.01 g SrCl_2_.6H_2_O, 1 mL Na_2_WO_4_ 10 mM, 1 mL FeCl_3_ 25 mM, 1 g yeast extract, 4 g tryptone, and 1 mg resazurin. The medium was adjusted to pH 6.8, autoclaved, and then reduced with 0.5 g sodium sulphide before use. The medium with the addition of 5g/L S^0^ was described as TRMS. All cultivation tests were performed in triplicates using standard anaerobic techniques.

Sulfide measurements were performed using the methylene blue method [51]. Samples were diluted 1:1 with 5% ZnAc solution directly after sampling to precipitate all sulfides. The solution was stored at room temperature for at least 20 min in order to promote the precipitation of zinc sulfide. After color development, the concentration was measured at 670 nm. Demi-water was used as a blank. 

### 3.2. RNA Purification and Transcriptome Sequence

Triple cell cultures in the exponential growth phase supplemented with or without S^0^ were harvested and immediately resuspended in TRIzol (Invitrogen, Waltham, MA, USA) to obtain total RNA. The DNaseI (Takara, Shiga, Japon) treatment was performed at 37 °C for 1 h and further purified by choloroform extraction and ethanol precipitation. The concentration and quality of purified RNA was checked by agarose gel electrophoresis and a NanoDrop 1000 spectrophotometer (Thermo Scientific Inc., Waltham, MA, USA). The RNA-Seq and subsequent bioinformatics analysis were carried out by Majorbio Biotech Co. in Shanghai, China. An Illumina HiSeq^TM^ 2000 platform was applied for the sequencing. Reads for further analysis were collected from sequence data passing quality control. A sequencing quality assessment, including alignment statistics, sequencing randomness assessment, and distribution of reads in the reference genome (CP006019), was carried out. Reads were mapped to a reference genome using SOAP2 [52]. The RNA-Seq reads were deposited in GenBank with accession number SRR10749089, SRR10749090.

### 3.3. Data Processing and Analysis

Differentially transcribed genes were identified using a rigorous algorithm based on the method described previously [53]. The calculation of unigene transcription used the RPKM (reads per kb per million reads) method [16]. The calculated gene transcription profile can be used to directly compare gene transcription levels between samples. Genes with log2 (ratio RPKM) values >2.0 or <−2.0 and significance in chi-square tests (*p* = 0.05, with Bonferroni correction) were considered to be the differently expressed genes between the two samples.

The genes were assigned to specific Clusters of Orthologous Groups (COG) Database (2012 archaeal COGs) f unctional gene groups. The homology of genes was analyzed by OrthoDB version 9.1 [54]. Hydrogenase analysis was further analyzed by HydDB and PSI-Blast [55].

### 3.4. Determination of Protease Activity

The crude extracts from culture supernatant and cell sonicate were assayed for proteolytic activity using casein (Sigma-aldrich, St. Louis, MO, USA) as a substrate at 60 °C, 80 °C, and 90 °C. Aliquots (100 µL) of samples were added to 100 µL of 2% casein in Tris-HCl, pH 8, containing 0.2 M NaCl. Fractions were then incubated at 60 °C, 80 °C, 90 °C for 10 min. The reaction was terminated by adding 130 µL of 10% trichloroacetic acid (TCA) (Sinopharm, Shanghai, China) and centrifuged at 10,000× *g* for 10 min. The supernatant (100 µL) was mixed with 500 µL, 0.5 M Na_2_CO_3_ and 100 µL of Folin’s phenol solution (Solarbio, Beijing, China) in distilled water, kept for 20 min at 40 °C, and measured for absorption at 660 nm. The tyrosine standard graph was prepared for a concentration range of 10 to 60 ug/mL to determine protease activity. One enzyme activity unit was defined as the amount of enzyme that was required to hydrolyze casein to yield 1 µg of tyrosine in 1 min [56].

## 4. Conclusions

Sulfur metabolism is important for the energy flow of deep-sea hydrothermal vents [57]. Most of these microorganisms utilize elemental sulfur (S^0^) as a terminal electron acceptor and reduce it to H_2_S, including Thermococcales species [8]. To date, limited metabolic analyses were studied in *Palaeococcus* species. *Palaeococcus pacificus* DY20341^T^ was isolated from hydrothermal sediments (limited sulfur, lower temperature), which differed from other Thermococcales species from active chimneys. In this study, we determined the transcriptome of *Palaeococcus pacificus* DY20341^T^ when it was grown with or without elemental sulfur. The results provided insight into the possible mechanism of hyperthermophiles, how they adapt to the hydrothermal fields between inactive hydrothermal sediments and active hydrothermal chimneys.

In the culture with elemental sulfur, DY20341^T^ upregulated sulfur-reducing proteins, pyridine nucleotide-disulfide oxidoreductase (NPSOR), NAD(P)H sulfur oxidoreductase (Nsr), sulfide dehydrogenases (SuDHI), and membrane-bound hydrogenase (MBX) which can produce H_2_S (Table 2, Figure 2). It indicated that more ATP was generated based on the sulfur-reducing process. Essentially, the change in the redox state of cells leads to an influence on the transcriptional process of cells. Both NADP(H) and ferredoxin or flavodoxin are important cofactors to the central metabolism in *Palaeococcus pacificus* DY20341^T^. In sulfur-supplemented culture, ferredoxin is generated by GAPOR which is further used by MBX and MBH1. Interestingly, we found that elemental sulfur also induced the upregulation of metal ion transfer (FeII, FeⅢ-siderphore, nickel/cobalt). Further, with sulfur supply, iron–sulfur cluster synthesis was triggered. This process resulted in more iron–sulfur cluster-carrying protein (NBP35) and some iron–sulfur cluster-containing proteins (Sip A, SAM, CobQ, HYP) being highly induced (Table 1, Figure 2). Cobalamin was the upregulated produced cofactor in this condition. These iron–sulfur cluster-containing proteins may contribute to electron transfer and energy production, further helping to reduce sulfur. Sulfur addtion induced iron and nickel/cobalt transport of strain DY20341. As a result, metal ion induced SurR regulator(3/4) were found downregulated in response to metal which not only regulates hydrogenase (MBH) and NAD(P)H sulfur oxidoreductase (Nsr) but also controls other transcriptional regulators such as TrmB which further downregulates starch degradation (Mal-II, Mal-IV).

Without extra sulfur, the energy of strain DY20341^T^ depends on the organic compounds including proteinous compounds, starch, sugars, etc. The upregulation of flagella will help the strains move for more foods. We found that in the genome of DY20341^T^, diverse extracellular and intracellular hydrolases including protease, peptidase, amylase, and β-glycoside hydrolase exist. Also, peptide/amino acids transporters and maltose/sugar transporters were introduced to yield more substrates to use (Figure 2). Cofactor riboflavin and thiamin biosynthesis were promoted in TRM culture.

When the DY20341 strain swims away from the active chimney or adapts to the dying chimney, it will change to fermentation growth to promote the utilization of the organic matter. Then, with the succession of the elemental sulfur, it will trigger the sulfur-reducing metabolism and metal assimilation to obtain more energy, dependent on the formation of the proton motive force coupled with ATP generation. These studies found that *Palaeococcus* species have versatile metabolisms to use organic matter or elemental sulfur. Also, the sulfur reducing process is a bit different with *P. furious* and *T. onnurineus.* The genome and transcriptome analysis of *Pa. pacificus* DY20341^T^ will contribute to a better understanding of the adaptation of hyperthermophilic Thermococcales species between hydrothermal inactive sediments and active chimneys.

## Figures and Tables

**Figure 1 ijms-21-00368-f001:**
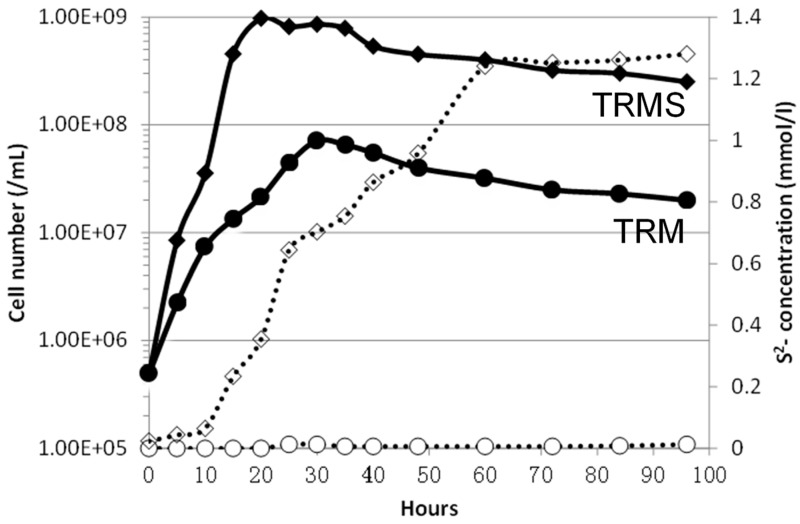
*Palaeococcus pacificus* DY20341^T^ cultured with or without elemental sulfur (S^0^). Data show the growth (solid line) and sulfide production (dotted line) of strain DY20341 cultured in Thermococcales rich medium with the elemental sulfur (◆ TRMS medium) and without (● TRM medium).

**Figure 2 ijms-21-00368-f002:**
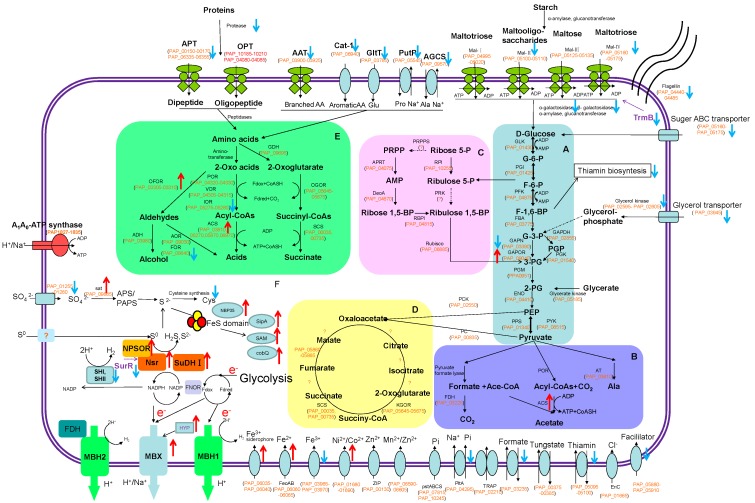
General metabolism and elemental sulfur-mediated cellular responses in *Palaeococcus pacificus* DY20341^T^. ↑: Upregulated genes; ↓: downregulated genes in the culture with S^0^. A: Modified Embden–Meyerhof glycolytic pathway; B: pyruvate degradation; C: pentose phosphate synthesis; D: pseudo-tricarboxylic acid cycle; E: amino acid degradation; F: sulfur metabolism and formation of proton motive force coupled with ATP generation. Abbreviations: POR: pyruvate:ferredoxin oxidoreductase; VOR: 2-ketoisovalerate:ferredoxin oxidoreductase; IOR: indolepyruvate:ferredoxin oxidoreductase; KGOR: 2-ketoglutarate:ferredoxin oxidoreductase; SCS: succinyl-CoA synthetase; ACS: acetyl-CoA synthetase; AOR: aldehyde:ferredoxin oxidoreductases; GLK: ADP-dependent glucokinase; PGI: phosphoglucose isomerase; GPI: glucose-6-phosphate isomerase; PFK: ADP-dependent phosphofructokinase; FBA: fructose-1,6-bisphosphate aldolase; GAPOR: glyceraldehyde-3-phosphate:ferredoxin oxidoreductase; GAPN: non-phosphorylating glyceraldehyde-3-phosphatedehydrogenase; PGM: phosphoglycerate mutase; PYK: pyruvate kinase; PGK: 3-phosphoglycerate kinase; GAPDH: glyceraldehyde-3-phosphate dehydrogenase; FBP: fructose-1,6-bisphos-phatase; FDH: formate dehydrogenase; Fd(ox), oxidized ferredoxin; Fd(red), reduced ferredoxin; Nsr: NADPH:S^0^ oxidoreductase; NPSOR: pyridine nucleotide-disulfide oxidoreductase; SipA, NBP35, Cmo, CobQ: iron–sulfur cluster-containing proteins; MBH: membrane-bound proton-reducing H_2_-evolving hydrogenase complexes; MBX: membrane-bound NADP-reducing hydrogenase; SH: cytoplasmic (Ni–Fe) hydrogenases; HYP: hydrogenase maturation proteins; SurR, TrmB: transcriptional regulators.

**Figure 3 ijms-21-00368-f003:**
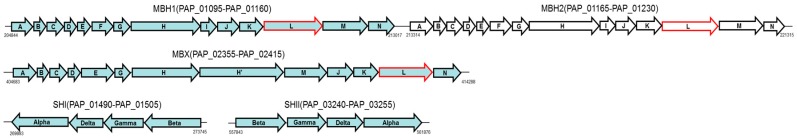
Hydrogenase operons in *Pa. pacificus* DY20341 including MBH1, MBH2, MBX, SHI, and SHII.The red arrow is the [Ni-Fe] catalytic subunit L to produce H_2_ or H_2_S.

**Table 1 ijms-21-00368-t001:** Top 10 upregulated or downregulated genes in *Pa. pacificus* grown with S^0^ (TRMS) compared to without it (TRM).

Type	Gene ID	TRMS- RPKM	TRM- RPKM	log_2_ (TRMS/TRM)	*p*-Value	Predicted Protein (Top Hit Species)
Up	PAP_06520	15,920	212	6.23	0	Fe-Mo cluster-binding protein, SipA (*Thermococcus gammatolerans* EJ3)
	PAP_06515	1980	101	4.29	0	Fe–Mo cofactor-binding protein (*Thermococcus barophilus* MP)
	PAP_06425	3373	253	3.74	0	Cobalamin biosynthesis protein, CobQ: iron–sulfur cluster binding domain (*Thermococcus barophilus* MP)
	PAP_01595	32,502	3010	3.43	0	Ni–Fe hydrogenase metallocenter assembly protein, HypF (*Thermococcus barophilus* MP)
	PAP_06430	2931	276	3.41	1.27 × 10^−^^244^	Cobalamin biosynthesis protein, CobQ: iron–sulfur cluster binding domain (*Thermococcus barophilus* MP)
	PAP_02400	1490	170	3.13	6.34 × 10^−^^191^	NADH dehydrogenase subunit B (*Thermococcus onnurineus* NA1)
	PAP_04045	1170	136	3.10	0	Hypothetical protein (*Thermococcus litoralis* DSM 5473)
	PAP_03165	8830	1083	3.03	1.83 × 10^−^^200^	Hypothetical protein (*Thermococcus kodakarensis* KOD)
	PAP_08665	1288	165	2.97	0	Hypothetical protein (*Pyrococcus yayanosii*)
	PAP_06035	3912	515	2.93	0	ABC-type iron (III)-siderophore transporter permease, ABC.FEV.P (*Thermococcus barophilus* MP)
Down	PAP_03140	7	97	−3.79	2.26 × 10^−^^22^	50S ribosomal protein L35Ae (*Thermococcus barophilus* MP)
	PAP_04135	21	223	−3.41	7.98 × 10^−46^	Hypothetical protein (*Thermococcus barophilus* MP)
	PAP_06725	12	103	−3.10	1.51 × 10^−^^22^	Cobalamin adenosyltransferase (*Thermococcus litoralis* DSM 5473)
	PAP_04130	36	293	−3.02	3.40 × 10^−54^	Diacetylchitobiose deacetylase (*Thermococcus barophilus* MP)
	PAP_05255	14	106	−2.92	4.46 × 10^−^^20^	Nucleotide-binding protein (*Thermococcus litoralis* DSM 5473)
	PAP_07560	30	222	−2.89	7.70 × 10^−40^	Hypothetical protein (*Ktedonobacter racemifer* DSM 44963)
	PAP_08745	87	631	−2.86	7.29 × 10^−9^	Cystathionine beta-synthase (*Pyrococcus* sp. NA2)
	PAP_04125	32	225	−2.81	2.03 × 10^−39^	Hydrolase (*Thermococcus barophilus* MP)
	PAP_04120	52	364	−2.80	1.53 × 10^−62^	Hypothetical protein (*Thermococcus barophilus* MP)
	PAP_06600	10	70	−2.80	2.49 × 10^−13^	Cro regulatory protein (*Thermococcus litoralis* DSM 5473)

**Table 2 ijms-21-00368-t002:** The expression difference of key genes in S^0^-reduction system between Thermococcales species.

Protein Function	Protein Name (Abbreviation)	*Pa. pacificus*	*P. furious*	*T. onnurineus*
Hydrogenase	MBH1	↑	↓	↓
	MBH2	-	↓	Does not exist
	MBX	↑	↑	↑
	SHI	↓	↓	↓
	SHII	-	↓	↓
Sulfur-reducing metabolism	SuDHI	↑	↑	↑
	SuDHII	Does not exist	Does not exist	↑
	NPSOR	↑	Does not exist	↑
	Nsr	↑	↑	↑
Iron-sulfur metabolism	SipA	↑	↑	↑
	SipB	Does not exist	Does not exist	↑
	FeoB	↑	↑	↑
	SufB	↓	↑	↑
	Mrp/NBP35	↑	N.D.	↑
Regulator	SurR	↓	↓	↓
Superoxide-reducing system	NROR	↑	N.D.	↓
	SOR	-	N.D.	↓
	Rr	-	N.D.	↓
ATP synthesis	ATP synthase	-	↑	↑

↑: Upregulated during growth on S^0^; ↓: downregulated during growth on S^0^; -: no obvious differential expression; Does not exist: no homolog found; N.D., not determined.

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
