# Peer review of "Metabolic Adaptation to Sulfur of Hyperthermophilic Palaeococcus pacificus DY20341T from Deep-Sea Hydrothermal Sediments"

_ijms, 2020, doi:10.3390/ijms21010368_

Round 1

Reviewer 1 Report

   The authors have been very responsive to raised concerns. Two minor last comments are:

In Figure 2, “sugar ABC transporter” and “glycerol transporter” are not correctly positioned. They should be located at in the membrane.

Please picture the sulfate transporter (ABC transporter; PAP_01255-01260) like other transporters. For instance, a green or blue elliptical shape in the membrane.

Author Response

Response to Reviewer 1 Comments

Reviewer 1

The authors have been very responsive to raised concerns. Two minor last comments are:

In Figure 2, “sugar ABC transporter” and “glycerol transporter” are not correctly positioned. They should be located at in the membrane.

Please picture the sulfate transporter (ABC transporter; PAP_01255-01260) like other transporters. For instance, a green or blue elliptical shape in the membrane.

Response: Thanks for your advice. We modified them all.

Reviewer 2 Report

This research paper is based on a very detailed analysis of the transcriptome of P. pacificus in response to medium supplemented with sulphur. The paper is very interesting, and the results could be very significant for the research community focused on extremophiles. However, as it is, the text is difficult to follow, the figures are presented in a very chaotic way, and the English language used in the manuscript should be professionally edited. Some specific points:

Line 14: What do you mean by “deployed”?

Line 15: Pa.pacificus should be P. pacificus, this should be changed throughout the text.

There are many sentences in the text that should be edited because they are very difficult to read, for instance: lines 24-26, 44, 49-50, etc.

Figure 1 is referred in the text as figure A1, please change that. In addition, the quality of figure 1 is quite low, and the figure caption is difficult to follow. Please, add a figure legend to make this graph self-explanatory.

Supplementary materials should be transferred to a separate file, and their name should differ from the main figures and tables. For instance, Figure 1 in the supplementary should be Figure S1, Tables A1 and A2 should be Tables S1 and S2, etc.

Section 2.3 is quite difficult to follow and with very little analysis of the biological significance of the results. This should be edited to guide the reader through the data presented in this part of the manuscript. On the other hand, the discussion should focus on the most important findings to present the big picture.

Line 277, this should be Figure 3, please check all figure names and figure captions, it is very difficult to follow your narrative argument.

Author Response

Reviewer 2

This research paper is based on a very detailed analysis of the transcriptome of P. pacificus in response to medium supplemented with sulphur. The paper is very interesting, and the results could be very significant for the research community focused on extremophiles. However, as it is, the text is difficult to follow, the figures are presented in a very chaotic way, and the English language used in the manuscript should be professionally edited. Some specific points:

Line 14: What do you mean by “deployed”?

Response: We modified “deployed” to “performed”.

Line 15: Pa.pacificus should be P. pacificus, this should be changed throughout the text.

Response:  Pa. stand for Palaeococcus genus; P stand for Pyrococcus genus; To avoid confusing these two genus, we suggest that Pa.pacificus  is better.

There are many sentences in the text that should be edited because they are very difficult to read, for instance: lines 24-26, 44, 49-50, etc.

Response: Thanks for your advice. We had a MDPI English Editing Service to improve it.

Figure 1 is referred in the text as figure A1, please change that. In addition, the quality of figure 1 is quite low, and the figure caption is difficult to follow. Please, add a figure legend to make this graph self-explanatory.

Response: We modified the mistake, and changed it to a better quality fig.

Supplementary materials should be transferred to a separate file, and their name should differ from the main figures and tables. For instance, Figure 1 in the supplementary should be Figure S1, Tables A1 and A2 should be Tables S1 and S2, etc.

Response: Thanks for your advice. We modified it and recheck the figure and table in the manuscript.

Section 2.3 is quite difficult to follow and with very little analysis of the biological significance of the results. This should be edited to guide the reader through the data presented in this part of the manuscript. On the other hand, the discussion should focus on the most important findings to present the big picture.

Response: Thanks for your advice. We added more analysis highlighted yellow in 2.3.Further, we add more discussion in 4.; We also improve the figure 2 .

Line 277, this should be Figure 3, please check all figure names and figure captions, it is very difficult to follow your narrative argument.

Response: Thanks for your advice.

Round 2

Reviewer 2 Report

I hace no further comments.

This manuscript is a resubmission of an earlier submission. The following is a list of the peer review reports and author responses from that submission.

Round 1

Reviewer 1 Report

Xiang Zeng and colleagues in this paper described the genome of a new isolate Palaeococcus pacificus and it’s metabolic adaptation to the use of elemental Sulfur as electron sink during a “mixotrophic” life style.

The article is well arranged although there are several aspects to be checked.

First of all, all the data were generated by a transcriptome experiment and only two conditions were tested: presence and absence of elemental sulfur that Pa. Pacificus is able to use as the electron acceptor.

In the material and methods, the statistical analysis used to discriminate between up and downregulation is completely missing.

Line 103-104 not clear

With just three replicate and without specified the method used to statistical asses the different expression (method hierarchical Bayes estimation of generalised linear mixed-effects model (Choi et al., 2008) or Benjamini–Hochberg FDR correction (Love et al., 2014) or others and the P-value or false discovery rate used to discriminate, all the results became very difficult to interpretation.

Where the data for transcriptome were deposited? Are they accessible?

Although Pa. Pacificus is described as hyperthermophilic piezophilic anaerobic archaeon, and in chapter 2.3.10 authors describe possible adaptation “against high temperature, high pressure, and oxygen”.

Line 388 the entire paragraph

If the variable was only the presence or absence of S0, how do the authors get these conclusions?

Line 72

growth temperature is not specified

Line 140-146

Do you have proof to asses that? Are you planning to arrange an experiment to test mixo-or-autho-troph ability of Pa. Pacificus.

Table 2

It is too long to put in the article, could be moved in supplementary and leave in the article only the main enzymes or a summary of the pathways up and dow-regulated.

Line 164-168

why save energy in the presence of elemental sulfur if it grows faster in this condition?

Line 288-295 330-336

In many parts in the text authors used obviously in different contexts as the data described are a direct consequence of the experiment performed here. Are you sure if is it correct? Every assertion are so easy to understand? Are the authors planing to perform more tests to prove these findings?

Line 302

insert reference for this assessment.

Line 359-360

rephrase the sentence

line 395

which link between the presence of sulfur and heat shock regulatory proteins? Why obviously??

line 413-414

could the high temperature (although not specified) mitigate the effect of high pressure and so Pa. pacificus does not show “thermo-pesophile” aminoacids?

Line 412

pacificens of pacificus?

Line 423

osmoregulation of Pa.pacificus during anaerobiosis?

Line 463-465

comment or what?

Line 490-497

New experiments (with media at low organic substrate concentration) could be useful to get inside to the metabolic potential of Pa. Pacificus. ex. Asses the ability to fix inorganic carbon …..

Reviewer 2 Report

In this study, the authors showed the genome sequence and performed transcriptomic analysis of Palaeococcus pacificus DY20341T with or without elemental sulfur by RNAseq. The data provided by transcriptional analysis is a new, interesting and important finding. However, the genome sequencing of Palaeococcus pacificus was completed and the genome data was reported in author’s previous paper. Therefore, author should clarify the data which were already reported. Furthermore, the transcriptome data generated by RNAseq (Table 2) need to be backed-up by other type of analyses, such as, biochemical experiments.

The key points are described below.

Major comment 1.

The genome sequence of Palaeococcus pacificus was already reported in author’s previous paper (Zeng X, Jebbar M, Shao Z. Complete Genome Sequence of Hyperthermophilic Piezophilic Archaeon Palaeococcus pacificus DY20341T, Isolated from Deep-Sea Hydrothermal Sediments. Genome Announc. 2015 Sep 17;3(5). pii: e01080-15). In the present text, they should carefully treat the genome data, which part is already published, and which part is not. It is also strange that the authors did not refer their report:

Lines 83-84. The major content in this sentence regarding the genome size (1,859,370 hp) and G + C content (43.04 %) of Palaeococcus pacificus was already revealed in their previous paper. If authors use these data, they should refer this paper correctly. Lines 13, 66, 472. These sentences would mislead readers. Because the genome sequence of pacificus was already reported, they should refer the previous paper and carefully describe about the previous findings. Similar reason to the comments 1-1 and 1-2, section “3.2. Genome sequence” should be removed from Materials and Methods. Figure 2 (Circular View of the Pa. pacificus genome) is probably generated from the data obtained from the previous paper. If so, the authors should explain in the literature. If not, they should clarify which part is already published.

Major comment 2.

The authors compare their results with those of other Thermococci species in various places in the text. Nevertheless, in some places, references are missing; for example, lines 163, 199, 207, 219, 230, 250, 299. Please refer previous papers correctly.

Major comment 3.

Table 2; please include the names of the top Blast Hit prokaryote and eukaryote with its identity and its PE score.

Major comment 4.

Biochemical evidence supporting the transcriptome data should be provided to make the finding solid.

Minor comments.

Lines 281-282, 284: sulfate adenylyltransferase and ATP sulfurylase are same enzyme (EC 2.7.7.4).

In Figure 2 (General metabolism and elemental sulfur-mediated cellular responses in Palaeococcus pacificus DY20341T): how do the archaea obtain sulfate from external milieu?

Line 47: reference 10 should be cited.

In KEGG and NCBI databases, alpha and beta subunits of Pa. pacificus fumarate hydratase (PAP_05860 and PAP_05865) (WP_048165115 and WP_048165116) are annotated. Please include these enzymes in Figure 2 and the text.

Line 143: Please show evidence for annotating alpha and beta subunits of succinyl-CoA synthetase (SCS, PAP_00035, PAP_00735).

In Thermococcus kodakaraensis, succinyl-CoA synthetase is indicated as a member of the enzymes involved in glutamate catabolism (Shikata K et al., JBC 282, 37, 26963–26970, 2007). Please refer this paper and discuss with your data of incomplete TCA cycle, which includes succinyl-CoA synthetase.

There are two Figure 1 and two Figure 2.

Round 2

Reviewer 2 Report

Comments to the authors.

In the revised version, the authors attempted to address all my concerns raised and revised satisfactorily some issues among them. However, there are still several points to which the authors should carefully address. Furthermore, as the authors agree with, the biochemical experiments need to be done to validate the transcriptome data generated by RNAseq.

The key points are described below.

Major comment 1.

Line 13; we report the genome sequence and 12 transcriptional analysis of Pa. pacificus DY20341T with/without elemental sulfur as electron acceptor.

Line 512. In this study, we determined the genome and transcriptome of Palaeococcus pacificus DY20341T, when it was 514 grown with or without elemental sulfur.

As I pointed out in the comments on the original version, the above sentences would mislead readers as if it was a new finding. Because the genome sequence of Pa. pacificus was already reported (ref 15), the authors should carefully treat and describe about the previous findings.

Major comment 2.

“Supplemental Figure 1A (Circular View of the Pa. pacificus genome)”, in which the data would be based on the genome sequence in the previous paper (ref 15), is not quoted in the text. As I commented on the original version, the authors should quote and explain for this supplemental Figure 1A in the text.

Major comment 3

As the authors agree in “Author response 4”, please add some biochemical data supporting the result of transcriptome analysis, for example, “protease activity test”. 

Minor comments.

In Figure 2 (General metabolism and elemental sulfur-mediated cellular responses in Palaeococcus pacificus DY20341T), please add hypothetical transporter on cellular membrane on the route of sulfate acquisition.

In Figure 2, the position of fumarate hydratase (PAP_05860 and PAP_05865) (WP_048165115 and WP_048165116) is not correct. Fumarate hydratase is the enzyme that catalyze “Malate <=> Fumarate + H2O”. The authors put “PAP_05860 and PAP_05865” on the position of succinate dehydrogenase.

Figure 2 is not quoted in the text.

Line 114; is reference No. 15 correct?
